# Effect of Azithromycin on Exacerbations in Asthma Patients with Obesity: Protocol for a Multi-Center, Prospective, Single-Arm Intervention Study

**DOI:** 10.3390/ijerph20031861

**Published:** 2023-01-19

**Authors:** Hiroki Tashiro, Koichiro Takahashi, Masaru Uchida, Yuki Kurihara, Hironori Sadamatsu, Ayako Takamori, Shinya Kimura, Naoko Sueoka-Aragane

**Affiliations:** 1Division of Hematology, Respiratory Medicine and Oncology, Department of Internal Medicine, Faculty of Medicine, Saga University, Saga 8498501, Japan; 2TARGET Investigator Group, Saga 8498501, Japan; 3Division of Internal Medicine, Japan Community Health Care Organization Saga Central Hospital, Saga 8498522, Japan; 4Division of Respiratory Medicine, Saga Prefectural Medical Center Koseikan, Saga 8408571, Japan; 5Clinical Research Center, Saga University Hospital, Saga 8498501, Japan

**Keywords:** asthma, obesity, azithromycin, microbiome

## Abstract

Introduction: Obesity is associated with severe asthma, but no specific treatment has been established. The gut microbiome is increasingly recognized as a crucial factor, but specific treatments focused on the gut microbiome have not been established. Recently, azithromycin has been found to have the capacity to attenuate exacerbations, a characteristic of severe asthma. The effect of azithromycin on obesity-induced severe asthma is not understood. Methods: The purpose of the present study is to clarify the effect of azithromycin on exacerbations in asthmatic patients with obesity. To explore the mechanism, the gut microbiome, metabolites of microbes such as short-chain fatty acids, and blood inflammatory cytokines will be analyzed to evaluate the correlation with the effect of azithromycin on exacerbations in obesity-induced severe asthma. A multi-center, prospective, single-arm intervention study is planned. Discussion: The present study will allow us to evaluate the effect of azithromycin on exacerbations, particularly in asthma patients with obesity, and explore biomarkers, targeting molecules including the gut microbiome, which are correlated with decreased exacerbations. The present results could contribute to identifying new therapeutic prospects and targeted microbes or molecules associated with severe clinical characteristics in asthmatic patients with obesity. Trial registration: This study has been registered as a prospective study with the University Hospital Medical Information Network (UMIN0000484389) and the Japan Registry of Clinical Trials (jRCTs071220023).

## 1. Introduction

Asthma is a common disease, and the majority of patients are well-controlled by treatment, including inhaled corticosteroids and bronchodilators, which are pivotal treatments for asthma [1,2]. However, approximately 5 to 10% of the patients are resistant to the standard treatments, and they are considered to have severe asthma [3]. Clinical phenotypes of severe asthma are heterogeneous, but recognized as symptomatic, with reduced pulmonary function, higher airway inflammation, and frequent exacerbations [4,5]. Exacerbation is one of the important events in severe asthma that is related to higher mortality, emergency hospital visits, increased medical costs, and lower quality of life [6,7,8]. Thus, better control exacerbations by treatment, along with exploring the severity mechanisms, are increasingly required. Recently, several biologics targeting molecules associated with eosinophilic inflammation such as interleukin (IL)-4, IL-5, IL-13, and immunoglobulin (Ig) E, called type 2 cytokines, have become available, and they have the capacity to provide control for some patients with severe asthma, especially those characterized by eosinophilic airway inflammation [9,10]. However, their efficacy is insufficient for patients with less or non-type 2 inflammation [11]. For such severe asthma patients without eosinophilic airway inflammation, exploring novel treatments and therapeutic targets is needed.

There is increasing evidence that overweight and obesity are associated with the severity of asthma. For example, overweight increases the asthma exacerbation rate [12], and obesity reduces the efficacy of the standard treatments including corticosteroids [13,14,15]. In addition, obesity accounts for approximately 60% of the general severe asthma population in the United States, which indicates that obesity is a major phenotype of severe asthma [16]. Unfortunately, this form of asthma does not have any specific and effective treatment other than weight reduction. Indeed, previous reports showed that diet- and/or exercise-induced weight loss improved asthma control, lung function, and asthma-related quality of life [17,18,19]. As mechanisms, decreased exhaled nitric oxide and improved allergic airway inflammation through regulatory T-cell function and neutrophilic inflammation were considered [19,20,21]. Because asthma with obesity includes more patients with clinical characteristics of less or non-eosinophilic inflammation than lean asthma [22], it is necessary to clarify the severity mechanisms of obesity-induced severe asthma as a treatment target and identify effective treatments focused on less eosinophilic inflammation.

The gut microbiome is recognized as essential in the severity mechanisms of obesity-induced asthma [23]. For example, specific microbes in the gut, especially the level of *Akkermansia muciniphila*, were found to be correlated with severity in obese asthmatic patients [24], and we also showed that the gut microbiome directly affected obesity-induced exacerbations of non-eosinophilic airway inflammation and airway hyper-responsiveness in vivo [25]. In addition, specific interventions for the gut microbiome, including prebiotics and antibiotics, attenuate exacerbations [25,26,27,28], indicating that the gut microbiome is a pivotal mechanistic target for obesity-induced severe asthma. In terms of the severity mechanisms via the gut microbiome, several studies focusing especially on body weight reduction by bariatric surgery, including Roux-en-Y gastric bypass and sleeve gastrectomy, as effective treatment for obesity-induced asthma have been reported [29]. For example, researchers have shown that bariatric surgery improved asthma control and airway hyper-responsiveness [30,31,32]. Importantly, bariatric surgery also has the capacity to change gut microbiome composition, including *Akkermansia muciniphila* and others [33,34,35]. Recently, the AMAZES study, a randomized, double-blinded, placebo-controlled trial, showed that azithromycin, which is one of the macrolide antibiotics, significantly reduced asthma exacerbations, and, importantly, the effects were significant for patients whether or not their inflammation status was eosinophilic [36]. Notably, azithromycin also has the capacity to alter the human gut microbiome [37] and attenuate airway inflammation in vivo through gut microbiome manipulation [38]. These data suggest that azithromycin has the possibility to reduce exacerbations of obesity-induced severe asthma with alterations of the gut microbiome, even though the clinical phenotype is mainly less or non-eosinophilic inflammation.

The present study is a multi-center, prospective, single-arm intervention study involving treatment with azithromycin for 48 weeks to clarify whether such treatment reduces exacerbations of obesity-induced severe asthma. To explore the mechanism, the gut microbiome, metabolites produced by microbes such as short-chain fatty acids, and blood inflammatory cytokines will be assessed to evaluate their correlation with the reduction in exacerbations by azithromycin in obesity-induced severe asthma. Importantly, the present study is designed as a single-arm intervention considering the gut microbiome and biomarker diversity of individuals, which allows us to avoid individual diversity at baseline and explore specific microbes and biomarkers associated with clinical outcomes with a relatively small sample size, taking into account ethical concerns. Compared to the AMAZES study mentioned above, the study design, participants, intervention dose of azithromycin, and study endpoints are largely different. Briefly, the present study is planned as a single-arm intervention study compared to a randomized, double-blinded, placebo-controlled trial in the AMAZES study because of the sample size calculation considering gut microbiome diversity of individuals set as secondary endpoints. As study participants, the present study targets Japanese asthmatic patients with obesity (BMI > 25 kg/m^2^) compared to asthma patients recruited in Australia and New Zealand regardless of their BMI in the AMAZES study. Considering efficacy and safety, the intervention dose of azithromycin is 250 mg per day in the present study compared to 500 mg, 3 times per week in the AMAZES study. Finally, the present study focuses on the gut microbiome and its metabolism as study endpoints. The present results will contribute to developing new therapies for obesity-induced severe asthma and identify targeted microbes or molecules associated with the regulation of asthma with obesity.

## 2. Materials and Methods

### 2.1. Study and Setting

The present study is a multi-center, open-label, prospective, single-arm intervention study. The duration of the intervention is 48 weeks. All patients with asthma are to be diagnosed by an expert pulmonary physician referring to the Global Initiative for Asthma (GINA) guidelines [39]. Briefly, the patients show asthma-related symptoms such as cough, sputum, shortness of breath, and/or wheezes, and clinical improvement, including the symptoms and/or forced expiratory volume in 1 s at more than 200 mL or 12% by treatment for asthma, including short-acting beta 2 agonists (SABAs) and inhaled corticosteroids. The body mass index (BMI) is calculated as the body mass divided by the square of the height (kg/m^2^). Obesity is defined as a BMI greater than 25 kg/m^2^, referring to the Japanese criterion [40]. In the present study, to focus on the effect of azithromycin on exacerbations in obesity-induced asthma patients, patients who have experienced one or more episodes of moderate or severe exacerbation in the previous year will be recruited according to the sample size calculation. Moderate exacerbations are defined as events that require a temporary or a permanent increase in the dose of inhaled corticosteroid, administration of antibiotics considering exacerbation of asthma-related symptoms, increased use of SABA for 2 or more consecutive days, and emergency visits related to asthma exacerbations without administration of systemic corticosteroids, referring to previous reports [36]. Severe exacerbations are defined as events that require administration of more than 10 mg of prednisolone for 3 consecutive days, hospitalization associated with the asthma exacerbation, and emergency visits related to asthma exacerbations with systemic corticosteroid administration, referring to previous reports [36]. The azithromycin regimen is 250 mg daily for 48 weeks, based on previous reports. Briefly, 48 weeks of azithromycin 500 mg, 3 times per week, showed a significant attenuating effect for exacerbations of asthma, but with 35% of patients experiencing diarrhea and 7% of patients stopping treatment due to severe adverse effects [36]. Another trial showed that 24 weeks of azithromycin 250 mg, 3 times per week, showed no significant effect on exacerbations of asthma compared to placebo [41]. In terms of adverse effects, the blood concentration of azithromycin 250 mg daily was significantly lower, with less adverse effects, including digestive disturbance and hearing disorder, compared to 500 mg, 3 times per week [42]. In addition, Japanese patients with non-tuberculosis mycobacteria who were administered azithromycin 250 mg daily for more than 1 year, which is approved by Japanese medical insurance, did not experience adverse effects other than skin rash in 3.2% [43,44]. According to these data, azithromycin 250 mg daily for 48 weeks was selected considering efficacy, safety, and standardization related to seasonal changes.

The present study was designed as a single-arm intervention considering the gut microbiome diversity of individuals. The composition of the gut microbiome differs depending on individual eating habits, medical history, and environmental factors [45,46,47], which suggests that exploring specific microbes associated with clinical outcomes would be difficult when comparing placebo and intervention groups. Therefore, the gut microbiome along with other biomarkers will be compared before and after intervention in a single-arm intervention, which allows us to avoid individual diversity at baseline. As concomitant drugs, corticosteroids and antibiotics other than macrolide antibiotics are allowed to treat exacerbations and/or infections at baseline and during participation in the present study. Antacid and anti-diarrheal agents are also allowed for ethical reasons, especially considering the adverse effects of azithromycin.

### 2.2. Inclusion Criteria

Patients diagnosed with asthma, age older than 20 years and less than 75 years, with BMI greater than 25 kg/m^2^ will be included. All patients must have experienced one or more episodes of moderate or severe exacerbations in the previous year. Written informed consent for participation in the present study will be obtained from all patients.

### 2.3. Exclusion Criteria

Exclusion criteria are the following: aspartate aminotransferase or alanine aminotransferase greater than 100 IU/L; creatinine above 2.0 mg/dL; life-threatening arrhythmia or paroxysmal tachycardia with heart rate greater than 100 beats per minute; admitted for acute myocardial infarction or heart failure in the previous year; malignancy diagnosed in the previous 5 years, except for complete remission after treatment; macrolide antibiotic treatment in the previous 4 weeks; biologics including omalizumab, mepolizumab, benralizumab, and dupilumab in the previous 8 weeks at the start of azithromycin in the present study; QT interval requires correction for heart rate at more than 480 ms; pregnancy or breastfeeding; BMI less than 25 kg/m^2^; and ineligible for the present study as determined by the researchers.

### 2.4. Study Endpoints

The primary endpoint is reduction in the exacerbation rate after 48-week treatment with azithromycin compared to that before the intervention (Table 1). The asthma exacerbation rate is defined as the proportion of analyzed cases with asthma exacerbations after treatment. Secondary endpoints are differences in clinical and laboratory parameters between before and after azithromycin treatment, including the blood eosinophil ratio, eosinophil count, IgE, pulmonary functions such as vital capacity (VC), forced vital capacity (FVC), %VC, forced expiratory volume in one second (FEV_1_), FEV_1_%, and %FEV_1_, and weight, BMI, level of fractional exhaled nitric oxide (FeNO), and scores on the asthma control test. Other secondary endpoints are to clarify the characteristics of the frequency of asthma exacerbations in improving/non-improving patients, and the diversity indices of the gut microbiome including the Chao 1, Shannon, and Simpson indices. For other exploratory endpoints, height, weight, BMI, and the tendency for decreased exacerbations will be assessed at 24 weeks of azithromycin treatment. In addition, biomarkers including cytokines, chemokines, short-chain fatty acids, metabolomics, and species of the gut microbiome will be compared between before and after azithromycin treatment and examined for relationships between changes and exacerbation at 48 weeks. The effects of sex and age attenuating the effect of azithromycin on exacerbations will also be assessed, because sex and age have been shown to have great impacts on the incidence and pathophysiology of asthma and obesity [26,48,49,50].

### 2.5. Clinical Data to Be Collected, including Adverse Events

The present study consists of 7 visits: before intervention, at azithromycin initiation, 4 weeks later, 12 weeks later, 24 weeks later, 36 weeks later, and 48 weeks later. Informed consent will be obtained at visit 1 with recording of clinical information including treatment contexts for asthma, smoking index, exacerbation times in the previous year, and comorbidities. Physical measurements, electrocardiogram, and blood examination will be evaluated at all visits except for visit 1 to monitor adverse effects of azithromycin. Pulmonary function testing, FeNO, and asthma control tests are to be performed at visits 2, 5, and 7. Biomarker analyses of blood, including multiplex assay of cytokines and chemokines (Eve Technologies, Calgary, AB, Canada), short-chain fatty acids, and metabolomics analysis (Kyusyu Pro Search, Fukuoka, Japan) are to be performed at visit 2 and visit 7. Gut microbiome analysis with 16s rRNA Sequencing (Massachusetts Host-Microbiome Center at the Brigham and Women’s Hospital, Boston, MA, USA) will be performed at visit 1 and visit 7. In terms of adverse effects of azithromycin, if the participants have any symptoms, physical and laboratory abnormalities of grade 2 level associated with azithromycin, the dose will be reduced to 250 mg 3 times per week. If the participants have any symptoms, physical and laboratory abnormalities of grade 3 level associated with azithromycin, administration of azithromycin will be terminated immediately.

### 2.6. Sample Size

A previous study reported that the incidence rate ratio of asthma exacerbations for azithromycin compared to placebo was approximately 0.6 [36], but the current participants have obesity, which is estimated to reduce the attenuation effect due to the pharmacological differences between lean and obese individuals [51,52]. Thus, we anticipate an incidence rate ratio of 0.9 with azithromycin and set the expected exacerbation rate after intervention at 90%, with the exacerbation rate before the intervention set at 100%. Based on the normal approximation of the binomial distribution with a significance level of 5% and power of 90%, 44 patients are required. Thus, the target sample size was set at 50 patients considering dropouts.

### 2.7. Gut Microbiome and Antibiotic Resistance Gene Analysis

For 16s rRNA sequencing for gut microbiome analysis, stool DNA will be isolated using QIAamp DNA Mini Kit (Qiagen, Germantown, MD, USA). A multiplex amplicon library converting the 16S rDNA gene V4 region will be generated from the DNA on the MiSeq machine (Illumina, San Diego, CA, USA). Microbial analysis will be performed using Qiime software with the Greengenes 99 database on the Nephele pipeline supported by the National Institutes of Health. The software will be used to calculate and generate alpha and beta diversities. For analysis of antibiotic resistance genes, tet(W), mel, msr(E), tet(M), erm(F), mef, and erm(B), which have been reported as genes related to antibiotic resistance [53], they will be examined by quantitative polymerase chain reaction using stool DNA.

### 2.8. Statistical Analysis

The primary and secondary endpoints and safety will be assessed in all patients completing 48 weeks of treatment. Then, patients without necessary exacerbation data for analysis or who withdrew consent before treatment will be excluded. For the primary endpoint, the asthma exacerbation rate at 48 weeks and its two-sided 95% confidence interval (CI) will be estimated with the normal approximation. The asthma exacerbation rate difference between pre- and post-treatment will also be tested binomially. For the clinical laboratory parameters and the diversity indices of the gut microbiome at pre-treatment and post-treatment as secondary endpoints, paired *t*-tests or McNemar’s tests will be used. The chi-squared test will be used to investigate the characteristics related to the frequency of asthma exacerbations. For other exploratory endpoints, height, weight, BMI, and tendency for an attenuation of the exacerbation rate at 24 weeks will be assessed by paired *t*-tests. Univariate analysis and multivariate analysis with a logistic regression model, adjusting BMI and body weight, will be performed to evaluate the relationships between the changes (improvement or no improvement in biomarker changes) and asthma exacerbation at 48 weeks.

## 3. Discussion

The present study focuses on asthma with obesity, which is one of the important forms of severe asthma [22], and this phenotype does not have effective treatments. Azithromycin, the intervention drug in the present study, is widely and continuously used for patients with non-tuberculosis mycobacterial infections [43], which guarantees its safety in terms of adverse effects. In addition, azithromycin has already been reported to have an attenuating effect on exacerbations in asthma patients regardless of their BMI [36], and, notably, it has the capacity to alter the gut microbiome as an antibiotic [37,38]. Thus, the present study will allow us to evaluate the attenuating effect on exacerbations, especially in asthma patients with obesity, and explore biomarkers, targeting molecules including the gut microbiome that have been correlated with decreased exacerbations. This is the first prospective intervention study to explore an effective drug for obesity-induced severe asthma focusing on the gut microbiome.

The gut microbiome is increasingly recognized as a mechanistic basis of obesity-associated conditions, including asthma. For example, the prevalence and pathophysiology of type 2 diabetes mellitus, metabolic syndrome, non-alcoholic fatty liver disease, and cardiovascular disease are associated with the gut microbiome [54,55,56]. Even in obesity itself, the gut microbiome has a huge impact, in that the relative abundances of the two dominant phylum-level bacteria, the *Bacteroidetes* and the *Firmicutes*, are significantly changed between obese and lean human volunteers [57]. In an asthma mouse model, we previously reported that the gut microbiome directly contributed to increasing ozone-induced airway hyper-responsiveness and neutrophilic airway inflammation [25,58]. Unfortunately, there are insufficient data about the specific association between obesity-induced severe asthma and the gut microbiome. However, body weight reduction by bariatric surgery is associated with the improvement of asthma pathophysiology and increased *Akkermansia muciniphila* in the gut [29,59,60]. These data remind us of the possibility of a mechanistic basis of obesity-induced severe asthma via the gut microbiome. Given these findings, focusing on the gut microbiome as a treatment target in the present study is consistent with the published data and could have an impact on the unmet needs in asthma with obesity.

Future prospects for gut-microbiome-targeted therapy are mainly considered to include fecal microbiota transplantation (FMT), prebiotics, and probiotics [23]. Indeed, the effect of FMT is widely recognized in patients with gastrointestinal disorders, including Clostridium difficile infection and inflammatory bowel diseases [61,62]. Prebiotics that affect the gut microbiome by dietary supplementation, such as dietary fiber, have potential to attenuate allergic and non-allergic airway responses in vivo [25,26,63]. Further, probiotics involving the administration of beneficial and specific taxa have been increasingly clarified, and *Akkermansia muciniphila*, *Turicibacter*, and *Lactobacillus* are the candidates related to asthma pathophysiology in vivo and in humans [23,24]. Unfortunately, we still do not have consistent clinical data associated with gut-microbiome-targeted therapies for severe asthma, regardless of BMI. Therefore, the focus is on the prudent use of antibiotics such as macrolides in the present study. Indeed, macrolides are recognized to have anti-inflammatory effects along with altering bacteria and improving asthma control, as well as suppressing sputum inflammatory cytokines [64]. We also reported that macrolides attenuate allergen-induced airway responses, especially those driven by type 2 and non-type 2 inflammation [27,28]. According to these data supported by the AMAZES study [36], the present study will contribute to exploring new therapeutic drugs for obesity-induced severe asthma as an unmet need in the current clinical setting. Notably, induction of antibiotic-resistant microbes would be taken into consideration because the present study plans to involve long-term administration of azithromycin. A sub-analysis of the AMAZES study showed that several genes related to antibiotic resistance were increased from sputum DNA [53]. We do not know the impact of antibiotic-resistant microbe induction in the present study because the participants and intervention dose of azithromycin are different compared to the AMAZES study. Thus, the focus will be on the analysis of genes related to antibiotic resistance.

The limitations of the present study are that there are no control arms of lean subjects and placebo. The phenotypes of severe asthma in lean and obese subjects are different, and standardization of inflammatory and background characteristics of these two groups would be difficult given the need for an increased sample size caused by the huge standard deviation. Furthermore, there is no placebo group because of the gut microbiome diversity of individuals, as mentioned in the methods, even though triggers of asthma exacerbation and other factors associated with gut microbiome changes might not be standardized between before and after the azithromycin intervention. Thus, a single-arm intervention has been planned for the present study. Another limitation is that the inclusion criterion of 20–75 years of age is a wide range, which might affect the results of the metabolomics analysis.

## 4. Conclusions

The present study will allow us to evaluate the effect of azithromycin on exacerbations, particularly in asthma patients with obesity, and explore biomarkers, targeting molecules including the gut microbiome, which are correlated with decreased exacerbations.

## Figures and Tables

**Table 1 ijerph-20-01861-t001:** Visit schedule of the present study.

	Visit 1	Visit 2	Visit 3	Visit 4	Visit 5	Visit 6	Visit 7
	Screening	AZM Start	4 Weeks	12 Weeks	24 Weeks	36 Weeks	48 Weeks
Informed consent	○						
Clinical information	○						
Medical examination	○	○	○	○	○	○	○
Physical measurement		○	○	○	○	○	○
Pulmonary function testing		○			○		○
FeNO		○			○		○
Asthma control test		○			○		○
Electrocardiogram		○	○	○	○	○	○
Blood examination		○	○	○	○	○	○
Biomarker analysis on blood		○					○
Gut microbiome analysis	○						○

AZM: azithromycin, FeNO: fractional exhaled nitric oxide. The circle indicated the measurement or obtaining of informed consent, information.

## Data Availability

The data of this study are stored and managed at the Saga University Clinical Research Center. The reader needs to contact the corresponding author regarding these requests.

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
