# Peer review of "Effect of Azithromycin on Exacerbations in Asthma Patients with Obesity: Protocol for a Multi-Center, Prospective, Single-Arm Intervention Study"

_ijerph, 2023, doi:10.3390/ijerph20031861_

Round 1
Reviewer 1 Report
The protocol manuscript titled "Effect of azithromycin on exacerbations in asthma patients with obesity: Protocol for a multi-center, prospective, single-arm, intervention study" is a interesting study and it is well written and clearly explained in all aspects. There are few changes needed, including the below comments, and importantly please proofread the whole document if all the citations are placed correctly.
Line 29, "with the regulation of asthma with obesity," does this refer to obese asthma patients? Please rewrite this sentence to be clear.
Line 82 - "Please include what are the major differences between your study protocol and the AMAZES study, because it will be helpful information for the readers as both the studies are about azithromycin and asthama."
Line 85 - This is not the correct reference - Reference 18 is not about azithromycin and asthma.
Line 86 - this is not correct reference.
Line 87 - This is not the correct reference. Please, It is suggested that you double-check that all the references are cited correctly throughout the document.
Line 151—Inclusion criteria for age are 20–75 years; the range is high; and will there be any effects on the study with the changes in metabolism that come with age? Including a statement explaining the reason for this choice would be helpful to the reader.
Line 197 - Gut microbiome analysis with 16s rRNA sequencing Please mention which sequencing method you are using.
Author Response
Response to reviewers for manuscript: ID: ijerph-2114492
Title: Effect of Azithromycin on Exacerbations in Asthma Patients with Obesity: Protocol for a Multi-Center, Prospective, Single Arm, Intervention Study
Journal: International Journal of Environmental Research and Public Health
We would like to thank the reviewers for their comments and suggestions. We believe that carefully considering and responding to these comments with statistician have improved the manuscript.
We revised the manuscript which was highlighted.
Reviewer 1
<Comment>
The protocol manuscript titled "Effect of azithromycin on exacerbations in asthma patients with obesity: Protocol for a multi-center, prospective, single-arm, intervention study" is a interesting study and it is well written and clearly explained in all aspects. There are few changes needed, including the below comments, and importantly please proofread the whole document if all the citations are placed correctly.
<Response>
Thank you very much for the comments. We followed the reviewer’s each suggestion and carefully double-checked the references whether those are correctly and appropriately cited.
<Comment>
Line 29, "with the regulation of asthma with obesity," does this refer to obese asthma patients? Please rewrite this sentence to be clear.
<Response>
Thank you very much for the pointing. We changed the uncleared expression of ‘with the regulation of asthma with obesity’ to ‘severe clinical characteristics in asthmatic patients with obesity’.
According to this, we revised the manuscript as below.
Page 1, Line 32
The present results could contribute to identifying new therapeutic prospects and targeted microbes or molecules associated with severe clinical characteristics in asthmatic patients with obesity.
<Comment>
Line 82 - "Please include what are the major differences between your study protocol and the AMAZES study, because it will be helpful information for the readers as both the studies are about azithromycin and asthma."
<Response>
I agree with the reviewer’s idea to show the major differences between present study and AMAZES study. The major differences between the 2 studies are study design, participants, intervention dose of azithromycin and study endpoints focusing on gut microbiome. Present study is planned as a single-arm intervention study comparing to a randomized, double-blinded, placebo-controlled trial in AMAZES study because of sample size calculation considering to the gut microbiome diversity of individuals set in secondary endpoints. For study participants, present study targets Japanese asthmatic patients with obesity whose BMI greater than 25 kg/m2 compared to asthma patients registered in Australia and New Zealand regardless of their BMI even though the specific races of them are not notified in AMZAES study. Considering to efficacy and safety, intervention dose of azithromycin is 250mg per day in present study compared to 500mg 3 times per week in AMAZES study. Finally, present study focuses on gut microbiome and the metabolism in study endpoints.
According to these, we revised the discussion part as below.
Page 3, Line 105
Compared to the AMAZES study mentioned above, the study design, participants, intervention dose of azithromycin, and study endpoints are largely different. Briefly, the present study is planned as a single-arm intervention study comparing to a randomized, double-blinded, placebo-controlled trial in the AMAZES study because of the sample size calculation considering gut microbiome diversity of individuals set as secondary endpoints. As study participants, the present study targets Japanese asthmatic patients with obesity (BMI > 25 kg/m2) compared to asthma patients recruited in Australia and New Zealand regardless of their BMI in the AMAZES study. Considering efficacy and safety, the intervention dose of azithromycin is 250 mg per day in the present study compared to 500 mg, 3 times per week in the AMAZES study. Finally, the present study focuses on the gut microbiome and its metabolism as study endpoints.
<Comment>
Line 85 - This is not the correct reference - Reference 18 is not about azithromycin and asthma.
<Response>
We apologize the incorrect references and we noticed that almost all of the reference data controlled by endnote are changed randomly. We double checked all of the references along with reference 18 about azithromycin and asthma.
<Comment>
Line 86 - this is not correct reference.
<Response>
As mentioned above, we double checked all of the references and revised.
<Comment>
Line 87 - This is not the correct reference. Please, It is suggested that you double-check that all the references are cited correctly throughout the document.
<Response>
We double checked all of the references and confirmed that everything is cited correctly and appropriately.
<Comment>
Line 151—Inclusion criteria for age are 20–75 years; the range is high; and will there be any effects on the study with the changes in metabolism that come with age? Including a statement explaining the reason for this choice would be helpful to the reader.
<Response>
As the reviewer pointed, the range of participants age in inclusion criteria of 20-75 years is high and it might contribute to results. Hence, we included age in study endpoints as reviewer 2 also suggested and clearly mentioned the risk of effect on metabolomics analysis related to the high range of age at 20-75 years.
Page 5, Line 201
The effects of sex and age attenuating the effect of azithromycin on exacerbations will also be assessed, because sex and age have been shown to have great impacts on the incidence and pathophysiology of asthma and obesity [26, 48-50].
Page 8, Line 525
Another limitation is that the inclusion criterion of 20-75 years of age is a wide range, which might affect the results of the metabolomics analysis.
<Comment>
Line 197 - Gut microbiome analysis with 16s rRNA sequencing Please mention which sequencing method you are using.
<Response>
For 16s rRNA sequencing for gut microbiome analysis, stool DNA is isolated using QIAamp DNA Mini Kit (Qiagen, Germantown, MD). A multiplex amplicon library converting the 16S rDNA gene V4 region is generated from the DNA on the MiSeq machine (Illumina, San Diego, CA). Microbial analysis is performed using Qiime software with Greengenes 99 database on the Nephele pipeline supported by National Institutes of Health. The software is used to calculate alpha and beta diversity.
We added this in method part as below.
Page 6, Line 235
Gut microbiome and antibiotic resistance gene analysis
For 16s rRNA sequencing for gut microbiome analysis, stool DNA will be isolated using QIAamp DNA Mini Kit (Qiagen, Germantown, MD). A multiplex amplicon li-brary converting the 16S rDNA gene V4 region will be generated from the DNA on the MiSeq machine (Illumina, San Diego, CA). Microbial analysis will be performed using Qiime software with the Greengenes 99 database on the Nephele pipeline supported by the National Institutes of Health. The software will be used to calculate and generate alpha and beta diversities. For analysis of antibiotic resistance genes, tet(W), mel, msr(E), tet(M), erm(F), mef, and erm(B), which have been reported as genes related to antibiotic resistance [53], will be examined by quantitative polymerase chain reaction using stool DNA.
Reviewer 2 Report
Effect of Azithromycin on Exacerbations in Asthma Patients 2 with Obesity: Protocol for a Multi-Center, Prospective, Single- 3 Arm, Intervention Study
4 Hiroki Tashiro 1,5, Koichiro Takahashi 1,5*, Masaru Uchida 2,5, Yuki Kurihara 1,5 , Hironori Sadamatsu 3,5, Ayako 5 Takamori 4 , Shinya Kimura 1 , Naoko Sueoka-Aragane 1
This is a designed protocol to be applied on the study of the effect of Azithromycin (AZM) on obese asthma in Japan. The manuscript is well written and easy to follow, parameters are well defined and well explained. Indeed, there is an increasing need for therapies specifically designed to the needs of obese patients and this could be a very interesting study. My comments are below:
1- The authors established by power calculation that their sample size is 44 patients. They increase it to 50 to include dropouts. This is less than 15% of dropouts in a study of 48 weeks. In my view this is a very optimistic number and unlikely to reflect the reality of such a long study. My advice would be to consider at least 20% of dropouts, 55 patients.
2- Obese asthma is a complex form of the disease and as the authors have explained, several studies suggest that it may be an entirely different cluster of patients, with predominant neutrophilic inflammation and higher number of exacerbations. In addition, sex differences in asthma are a factor widely accepted. In this regard, the literature suggest that the incidence of obesity is also higher in women than in men (Ng, M. et al., doi:10.1016/S0140-6736(14)60460-8 (2014), (Kanter, R. & Caballero. doi:10.3945/an.112.002063 (2012)). In addition, age of the patients seems to be important considering the peak age for the incidence of overweight and obesity for females has been described as closer to 60 years old in developed countries and 55 years of age in developing countries (Ng M. et al.). Despite the evidence, these two factors have not been included in this protocol, the authors have designed a very open inclusion criteria where sex and age are not considered in the analysis. In my view, this is important and without it, the analysis may not be accurate or complete.
3- As my final comment I would like to raise the issue of antibiotic resistance. Even though AZM show promising effects on asthma, it is an antibiotic, and this is fact cannot be ignored. There are studies in the literature showing that long term administration of AZM can lead to antibiotic resistance. Is there any evidence that 250mg of AZM daily will not have this side effect? The authors should not ignore this potential side effect of AZM in their discussion.
Author Response
Response to reviewers for manuscript: ID: ijerph-2114492
Title: Effect of Azithromycin on Exacerbations in Asthma Patients with Obesity: Protocol for a Multi-Center, Prospective, Single Arm, Intervention Study
Journal: International Journal of Environmental Research and Public Health
We would like to thank the reviewers for their comments and suggestions. We believe that carefully considering and responding to these comments with statistician have improved the manuscript.
We revised the manuscript which was highlighted.
Reviewer 2
<Comment>
This is a designed protocol to be applied on the study of the effect of Azithromycin (AZM) on obese asthma in Japan. The manuscript is well written and easy to follow, parameters are well defined and well explained. Indeed, there is an increasing need for therapies specifically designed to the needs of obese patients and this could be a very interesting study. My comments are below:
<Response>
Thank you very much for the reviewing. We carefully responded to specific comments with statistician which are described below.
<Comment>
1- The authors established by power calculation that their sample size is 44 patients. They increase it to 50 to include dropouts. This is less than 15% of dropouts in a study of 48 weeks. In my view this is a very optimistic number and unlikely to reflect the reality of such a long study. My advice would be to consider at least 20% of dropouts, 55 patients.
<Response>
Thank you very much for the advice and suggestion. We considered only 15% of dropouts even though the duration of intervention in present study is 48 weeks which might be risk of failure to archive primary endpoint as reviewer pointed. For AMAZES study, 45 patients were withdrawn including 15 patients with adverse events in 213 patients allocated to azithromycin which indicated that 30 patients (14%) were dropouts without adverse effect. Dose of azithromycin in present study is 250mg per day which expected significant low appearance of adverse effect referring to previous report (Kobayashi T. et al, Respir Investig 2021, 59, 212-7) which indicated no experience of adverse effect associated with cassation of azithromycin treatment in Japanese patients with non-tuberculosis mycobacteria. Additionally, considering to frequent visits as shown in Table 1 comparing to AMAZES study, we believe that 15% of dropouts might be sufficient.
<Comment>
2- Obese asthma is a complex form of the disease and as the authors have explained, several studies suggest that it may be an entirely different cluster of patients, with predominant neutrophilic inflammation and higher number of exacerbations. In addition, sex differences in asthma are a factor widely accepted. In this regard, the literature suggest that the incidence of obesity is also higher in women than in men (Ng, M. et al., doi:10.1016/S0140-6736(14)60460-8 (2014), (Kanter, R. & Caballero. doi:10.3945/an.112.002063 (2012)). In addition, age of the patients seems to be important considering the peak age for the incidence of overweight and obesity for females has been described as closer to 60 years old in developed countries and 55 years of age in developing countries (Ng M. et al.). Despite the evidence, these two factors have not been included in this protocol, the authors have designed a very open inclusion criteria where sex and age are not considered in the analysis. In my view, this is important and without it, the analysis may not be accurate or complete.
<Response>
We do agree with the idea focusing on sex and age differences in efficacy of azithromycin in asthmatic patients with obesity refereeing to previous reports indicated by reviewer. Hence, we included sex and age in exploratory endpoints and explained in method with citation of the references.
Page 5, Line 201
The effects of sex and age attenuating the effect of azithromycin on exacerbations will also be assessed, because sex and age have been shown to have great impacts on the incidence and pathophysiology of asthma and obesity [26, 48-50].
<Comment>
3- As my final comment I would like to raise the issue of antibiotic resistance. Even though AZM show promising effects on asthma, it is an antibiotic, and this is fact cannot be ignored. There are studies in the literature showing that long term administration of AZM can lead to antibiotic resistance. Is there any evidence that 250mg of AZM daily will not have this side effect? The authors should not ignore this potential side effect of AZM in their discussion.
<Response>
Thank you very much for the pointing. We also consider the importance of antibiotic resistance because present study involves long-term administration of azithromycin. Hence, we clearly mentioned the possibility of induction of azithromycin resistant bacteria in discussion part and included examination of gene related to antibiotics resistance including tet(W), mel, msr(E), tet(M), erm(F), mef, and erm(B) by quantitative polymerase chain reaction reported by previous report in stool DNA (Taylor SL. Et al, Am J Respir Crit Care Med. 2019 Aug 1;200(3):309-317.).
Page 6, Line 242
For analysis of antibiotic resistance genes, tet(W), mel, msr(E), tet(M), erm(F), mef, and erm(B), which have been reported as genes related to antibiotic resistance [53], will be examined by quantitative polymerase chain reaction using stool DNA.
Page 7, Line 309
Notably, induction of antibiotic-resistant microbes would be taken into consideration because the present study is planned to involve long-term administration of azithromycin. A sub-analysis of the AMAZES study showed that several genes related to antibiotic resistance were increased from sputum DNA [53]. We do not know the impact of antibiotic-resistant microbe induction in the present study because the participants and intervention dose of azithromycin are different compared to the AMAZES study. Thus, the focus will be on the analysis of genes related to antibiotic resistance.
Round 2
Reviewer 1 Report
All revisions are clearly indicated, and the right placement of your citations is evident.